# Synthesis, Characterization and Antimicrobial Activity of Zinc Oxide Nanoparticles against Selected Waterborne Bacterial and Yeast Pathogens

**DOI:** 10.3390/molecules27113532

**Published:** 2022-05-31

**Authors:** Michael John Klink, Neelan Laloo, Anny Leudjo Taka, Vusumzi Emmanuel Pakade, Mzimkhulu Ephraim Monapathi, Johannes Sekomeng Modise

**Affiliations:** Department of Biotechnology/Chemistry, Faculty of Applied and Computer Sciences, Vaal University of Technology, Vanderbijlpark 1900, South Africa; laloon@vut.ac.za (N.L.); lytany04@yahoo.fr (A.L.T.); vusumzip@vut.ac.za (V.E.P.); monapathimz@gmail.com (M.E.M.); joe@vut.ac.za (J.S.M.)

**Keywords:** zinc oxide, nanoparticles, bacterial, yeast, pathogens

## Abstract

The disinfection of wastewater using nanoparticles (NPs) has become a focal area of research in water treatment. In this study, zinc oxide (ZnO) NPs were synthesized using the microwave heating crystallization technique and characterized using transmission electron microscopy (TEM), Fourier transform infrared (FTIR) spectroscopy and X-ray diffraction (XRD). Qualitative well diffusion and quantitative minimum inhibitory concentration (MIC) tests were conducted to determine the antimicrobial activity of ZnO NPs against selected waterborne pathogenic microbes. FTIR spectral studies confirmed that the binding of urea with Zn occurs through Zn–O stretching. XRD confirmed the crystallized identity in a hexagonal ZnO wurtzite-type structure. The formation of zones of inhibition and low MIC values in the antimicrobial analysis were indicative of the effective antimicrobial activity of zinc oxide nanoparticles against the test microorganisms. The application of metallic nanoparticles in water treatment could curb the spread of waterborne microbial diseases.

## 1. Introduction

The world is facing serious water quality and quantity challenges. Millions of people are being confronted with a huge scarcity of water supply. The United Nations World Water Development Report, 2018 edition, predicted that, by 2050, 6 billion people will face water scarcity [1]. People will not have an adequate water supply due to increasing population and economic growth, which have resulted in increased water demand, decreasing available water resources and increasing the pollution of water resources. Poor water quality has increased the number of waterborne pathogens globally. Water-related diseases caused by these pathogenic organisms are the leading causes of death [2]. Humans are infected by these pathogens through drinking water and recreational activities. The proper and adequate treatment of water is necessary to reduce the risk of waterborne diseases, thus protecting public health [3].

In 2015, the member states of the United Nations established Sustainable Development Goals (SDGs) to create a global partnership among all countries in the world. SDG number 6 in the agenda stipulated that, by 2030, all people will have universal and equitable access to safe and affordable drinking water [4]. Due to ineffective treatment facilities, domestic and industrial wastewater effluents are frequently discharged into surface water sources, which are also the main drinking water resources [3]. The aforesaid target goal initiative was triggered by challenges facing water and wastewater resources to provide safe drinking water to communities, especially in developing countries. The water quality of global water resources is affected by increasing population growth, industrialization and climate change [5]. The resulting deteriorating water quality, increased water demand and decreased water supply threaten the achievement of SDG number 6. The provision of good water quality is essential to achieve SDGs for human health, food and water security.

The microbial contamination of water poses a great risk to human health [6]. Antibiotics are designed to limit or prevent the growth of microbes. However, pathogenic microbes are increasingly developing antibiotic resistance. Resistance results from continuous use, noncompliance and the misuse of antimicrobial agents [7]. Antimicrobial resistance is accompanied by a high global morbidity, mortality and high medical costs [8]. Thus, effective alternative antimicrobial strategies to control the growth of microbial populations are currently sought after [9]. 

The application of nanotechnology in water purification has been receiving great attention for treating persistent and emerging pathogenic contaminants in water [10]. Nanoparticles (NPs) are particles of matter with a typical nanoscale size of 1–100 nm in at least one dimension [11]. Their functional antibacterial activity stems from their inherent high surface-to-volume ratio due to their smaller particle sizes. These characteristics ensure close bonding interactions between the nanoparticles and microbial membranes [12]. The relatively large surface area and small size of NPs enable them to penetrate the cell membranes of pathogenic microorganisms, increasing their antibacterial effectiveness. Nanoparticles of smaller sizes have been found to exhibit greater toxicity on microorganisms [13]. Considered the next-generation antibiotics, NPs could limit the global crisis of emerging and increasing antimicrobial resistance in pathogenic microbes [14].

The commercial application of nanoparticles has gained considerable interest in the 21st century. The alarming increase in the application and integration of ZnO nanoparticles into agricultural, industrial and biomedical spheres has also attracted huge research interest [15,16]. According to Colon et al., 2006, and Padmavathy and Vijayaraghavan, 2008, ZnO NPs are non-toxic to human cells, harmful to microorganisms and possess excellent biocompatibility with human cells [17,18]. This qualifies their biomedical use as antimicrobial, anticancer, anti-diabetic and anti-inflammatory agents and in wound healing, drug/gene delivery, cell imaging and biosensors [15,19,20]. Due to their strong UV absorption properties, ZnO particles have also been integrated into personal care products, such as cosmetics, deodorants and sunscreens. ZnO nanoparticles have been used in industries that include concrete, coating, paint, rubber, photocatalysis, electronics and electrotechnology [16,21]. 

Previous studies have addressed the use of nanoparticles to impede the growth of some waterborne pathogens [22,23,24]. Their potential for treatment and microbial control in wastewater and drinking water environments warrants further research. The success of NPs would eventually reduce the mortality rate that is caused by the consumption of untreated microbially contaminated water. Zinc oxide is a metallic nanoparticle that has attracted huge research interest as an antibacterial agent due to its stability under harsh processing conditions and safety profile to humans and animals [25]. The present paper aims to synthesize and characterize zinc oxide nanoparticles and to determine their antibacterial activity against selected waterborne bacterial and yeast pathogens. 

## 2. Materials and Methods

### 2.1. Urea-Based Synthesis of Zinc Oxide Nanoparticles

The synthesis of ZnO NPs was carried out by dissolving 3.006 g of urea (CO(NH_2_)_2_) and 10.975 g of zinc acetate (Zn(CH_3_COO)_2_·2H_2_O) (99%, Merck, Kenilworth, NJ, USA) in 50 mL deionized water under constant agitation (500 rpm) [26]. The pH of the solution was adjusted to pH 12 by adding incremental volumes of 3 mol·L^−1^ KOH. The reaction contents were stirred for 15 min at ambient temperature. The solution was then heated at 100 °C for 4 h and left to cool at room temperature. Subsequently, the prepared aqueous solution was irradiated with a microwave synthesis instrument (LG-MS1040SM) for 2 and 8 min at 510 W. This enables the rapid and homogenous heating of the reaction mixture to the desire temperature, which saves time and energy. Finally, centrifugation was used to separate the resulting white precipitate, followed by washing with water and ethanol, and then it was dried for 5 h at 60 °C. ZnO NPs were prepared using the following ratios (Zn acetate: urea) (1:1, 1:2, 1:4, 2:1 μg·mL^−1^).

### 2.2. Characterization of the Synthesized Zinc Oxide Nanoparticles

A physical characterization of synthesized ZnO NPs was carried out to determine their optical and structural properties.

#### 2.2.1. UV–Visible Spectroscopy

The optical absorption spectrum of nanoparticles was obtained using a UV–Visible (UV–Vis) spectrophotometer (T80+ UV–Vis spectrometer). A total of 1 mg of the ZnO NP samples was dispersed in 1.5 mL of toluene reagent and sonicated for 30 min. The spectrum was recorded in the range of 200–600 nm [27].

#### 2.2.2. Fourier Transform Infrared Spectroscopy

An analysis of the ZnO NPs was carried out using Fourier Transform infrared spectroscopy (Perkin Elmer spectrum 400). ZnO NP samples were placed on a sample holder and tightly clamped. Samples were scanned (16 scans) from 400 to 4000 cm^−1^ at a resolution of 4 cm^−1^ [28,29]. 

#### 2.2.3. Scanning Electron Microscopy

A Vega 3 Tescan scanning electron microscope was used to determine the surface morphologies of the synthesized nanomaterials. For the analysis, a few mg of each sample was placed on a carbon tape. The samples were then gold-coated in order to make their surfaces conductive. Nanoparticle samples were loaded on a movable stage operated under a vacuum, and the sample surfaces were scanned by moving the electron-beam coils under the scanning electron microscope (SEM). The beam enabled the information about the defined area of the sample to be collected as images. It showed detailed images at high magnifications (up to 300,000×). Images were created without light waves (black and white) [30].

#### 2.2.4. X-ray Diffraction

The samples were analyzed on an X’Pert Pro XRD with a Co tube. The phases were identified using X’Pert Highscore plus software, PAN ICSD and an ICDD database. Samples were loaded on metal slides and placed inside the X-ray diffraction (XRD) analyzer. XRD patterns were collected between the angles 2θ of 20 and 80° at a scan rate of 2°/min using a diffractometer [31]. The data were subsequently collected and interpreted with graphs (with Joint Committee on Powder Diffraction Standards (JCPDS) card no. 36–1451). The sizes of the nanoparticles were calculated using the Debye–Scherer equation [32].

#### 2.2.5. Transmission Electron Microscopy

To identify the particle size and morphology of the prepared samples, transmission electron microscopy (TEM), a JEOL-JEM–2100 electron microscope, was used. Samples were dispersed in distilled water and sonicated. A drop of the sample was deposited onto a copper grid of TEM and observed under high magnification (300,000×) [33].

### 2.3. Antimicrobial Activity of ZnO Nanoparticles

Synthesized ZnO NPs were evaluated for their antimicrobial activity against the Gram-negative and -positive bacteria, namely, *Staphylococcus aureus* ATCC 25923, *Escherichia coli* ATCC O157, *Shigella sonnei* ATCC 25931 and *Salmonella enterica* ATCC 35664, and yeast *Candida albicans* ATCC 14053. Bacterial and yeast strains were grown in nutrient broth and potato dextrose broth, respectively, at 37 °C for 24 h. Shaking incubation was carried out at 150 rpm to reach a concentration of colony-forming units (CFU) of ∼10^6^ per/mL.

#### 2.3.1. Agar Well Diffusion Technique

The antimicrobial activities of the synthesized zinc oxide NPs were carried out using the well diffusion method [34]. Overnight cultures of the test organisms (100 µL) were spread on Mueller–Hinton (MH) (Biolab Merck, Germany) agar plates. Wells with diameters of about 6 mm were created aseptically using sterile micropipette tips on the agar plates. Twenty microliters of different synthesized zinc oxide nanoparticle ratios (Zn acetate: urea) (1:1, 1:2, 1:4, 2:1 μg·mL^−1^) were loaded in the MH agar wells. The positive controls used were a standard commercial antibiotic, neomycin (30 mg·mL^−1^), for bacterial isolates and amphotericin B (30 mg·mL^−1^) for yeasts, and they were added in one of the wells. The MH plates were incubated at 37 °C for 24 h. Any resulting clear zone (in mm) (zone of inhibition) around the wells was measured. The agar well diffusion experiment was performed in duplicate.

#### 2.3.2. Minimum Inhibitory Concentration (MIC) Assay

The inhibitory effect of the synthesized zinc oxide NPs on the growth of the test organisms was studied using the minimum inhibitory concentration (MIC) assay. The standard CLSI broth microdilution method was used to determine the MICs (NCCLS, 2002). The antimicrobial effectiveness of the synthesized zinc oxide nanoparticles was determined against a final microorganism concentration of 10^6^ colony-forming units mL^−1^. Disposable microtitration 96-well plates were used for the tests. MICs were determined by two-fold serial dilutions from prepared standard concentrations of zinc oxide nanoparticles: (A) 50 µg·mL^−1^, (B) 25 µg·mL^−1^, (C) 12.5 µg·mL^−1^, (D) 6.25 µg·mL^−1^, (E) 3.125 µg·mL^−1^, (F) 1.563 µg·mL^−1^, (G) 0.781 µg·mL^−1^ and (H) 0.391 µg·mL^−1^. The wells were inoculated with ten microliters of the overnight-grown test cultures and 0.02% resazurin dye. The negative control (no antibiotic or NPs added) and positive controls (neomycin (2 mg·mL^−1^) and amphotericin B (2 mg·mL^−1^)) were incorporated into the wells. The plates were incubated aerobically at 37 °C overnight. Afterward, the wells were observed for any color change. 

## 3. Results and Discussion

### 3.1. Synthesis of ZnO NPs

In the crystal growth process, first, tiny ZnO crystalline nuclei formed, and the nanoparticles of this oxide then precipitated by an increase in pH. This was due to the generation of NH_4_^+^ ions from NH_3_, which resulted from urea decomposition when the temperature rose. NH_4_^+^ ion formation is controlled by ammonia in water, and the hydrolysis of urea leads to a rise in pH due to the formation of hydroxyl or hydroxide groups. Urea hydrolysis progresses slowly, and the basic solution undergoes supersaturation of the zinc hydroxide species [35].

Additionally, the appropriate control of reaction time, temperature, solution pH and the choice of solvent type enables the production of pure oxide nanoparticles with desired particle sizes and shapes in a short time during crystallization by microwave heating [36]. Urea dissolves readily in water, forming carbon dioxide and ammonia when the hydrolysis reaction is aptly controlled. In this study, tiny ZnO crystalline nucleic particles formed during the initial phase of the crystal growth process. The NH_4_^+^ ions produced from the thermal decomposition of urea aided in the precipitation of ZnO particles by increasing the solution pH. The rise in solution pH is attributed to the formation of hydroxyl or hydroxide groups during the hydrolysis of urea and the subsequent reaction of ammonia with water. Thus, the solution became supersaturated with Zn(OH)_2_ species under basic conditions due to the slowness of the hydrolysis reaction of urea [35].

### 3.2. Characterization of the Synthesized ZnO Nanoparticles

To achieve the transition of unoccupied electronic states of the sample in UV–Visible spectroscopy, between the valence and conduction bands, the sample particles were illuminated with light. The spectra were recorded in absorption mode, and all samples were prepared by dissolving the nanoparticles in distilled water [37]. The absorption spectrum of the white crystal ZnO colloid particles produced from the urea reduction is shown in Figure 1. It can be observed that the particles are of spherical shape, which is also indicated by a surface plasmon absorption band with a maximum of 320 nm [33]. The emission peaks at 430 nm, 475 nm, 480 nm and 675 nm were accredited to the zinc vacancy, and the different deep level defect states originated from oxygen vacancies and/or the zinc interstitial vacancies [38,39].

The Fourier transform infrared spectra of ZnO NP nanopowders analyzed in the range of 1500–450 cm^−1^ are illustrated in Figure 2. Absorption bands at wavelengths 457 cm^−1^, 1063 cm^−1^, 1390 cm^−1^, 1602 cm^−1^ and 3423 cm^−1^ were observed in the FTIR spectrum of the as-prepared ZnO particles. The sharp peak positioned at 457 cm^−1^ is a characteristic of the presence of Zn–O stretching bonds. The presence of C=O, C–O and C–H vibrations is corroborated by the absorption bands appearing between 1700 and 600 cm^−1^. The other peaks are ascribed to the bending modes of the adsorbed water and O–H stretching vibrations [40].

Scanning electron microscopy is a very useful technique in characterizing the morphology of nanoparticles. The SEM results of ZnO NPs support the microcrystalline nature of the particles after calcinations with the least degree of agglomeration, as shown in Figure 3. The microwave heating process with a duration of 4 min yielded uniform ZnO NPs with a spherical shape. Thus, this shows that the microwave heating process is capable of producing uniformly shaped particles in a short analysis time.

The interface mobility and diffusivity in the medium enhanced by the MH process are credited for producing the uniform nanosized structural shapes of ZnO in a short time [26], whilst hydrothermal conditions are known to produce structures of different average particle distributions. Average particle diameters of 85 nm and 90 nm were reported for samples treated using the MH process for 8 and 2 min, respectively. 

The XRD patterns of the ZnO at different ratios are shown in Figure 4. The XRD pattern of the ZnO nanopowder at different ratios revealed that the sample under study exhibited high purity, as demonstrated by the single phase with no extra peaks. The sample was crystallized in a hexagonal structure [26]. The results reveal that microwave heating processing enhances the complete crystallization of ZnO NP at a low temperature and reduces processing time. All the observed peaks, (100), (002), (101), (102), (110), (103) and (112), can easily be identified in Bragg’s reflection associated with the hexagonal wurtzite phase of ZnO. The peak broadening in the XRD patterns clearly indicates the presence of small crystallites in the samples. The lattice constants a and c of all nanoparticles were calculated from the peak positions of the (100) and (002) planes, respectively. As a reference, the Joint Committee on Powder Diffraction Standards (JCPDS) card no. 36–1451 was used for peak identification, and it showed that the ZnO nanoparticles adopted a hexagonal ZnO wurtzite-type structure. The peaks’ broadness decreased continuously as the ratio of zinc acetate to urea increased, and this indicated an improved crystallinity and increased particle size [41].

The average crystallite size of the most intense XRD peak (the dominant phase) was calculated using the Scherrer Equation (1) below, and it was determined to be 85 nm.
D = kλ/βcosθ(1)
where D is the average crystallite size; k is a constant equal to 0.9; λ is the wavelength (nm) of copper Kα radiation, and its value is 0.154 nm; β is the FWHW of the peak of interest obtained by XRD; and Theta (θ) is the Bragg angle.

Transmission electron microscopy is a very useful technique in characterizing the size and shape of nanoparticles. Microstructural characterization studies were performed to determine the size of the ZnO NPs (Figure 5), as well as homogeneity and size distribution. The high-resolution transmission electron micrograph (HRTEM) shows clear platelets with round and hexagonal shapes, slightly agglomerated in a chain-like network (for different ratios and mixed-shaped particles).

The size of the particles and their distribution were estimated using the histograms shown in the insets of Figure 5a–d. These histograms were obtained by examining numerous frames of images. The average particle sizes as estimated from the TEM images are 100, 75, 40 and 33 nm for ZnO NP samples with ratios of 1:1, 1:2, 1:4 and 2:1, respectively. These results agree well with the XRD graph and particle size for using microwave heating to synthesize ZnO NPs.

### 3.3. Evaluation of Antimicrobial Activity

A qualitative agar well diffusion test was performed to screen ZnO NPs for their antimicrobial properties against yeast (*Candida albicans*), Gram-positive bacteria (*Staphylococcus aureus*) and Gram-negative bacteria (*Escherichia coli*, *Salmonella enterica* and *Shigella sonnei*) using the prepared ZnO NP ratios of 1:1, 1:2, 1:4 and 2:1 μg·mL^−1^. The growth-inhibiting effect of ZnO NPs was shown by the formation of clear zones on the Mueller–Hinton agar plates (Figure 6). The larger the zone of inhibition, the more sensitive the test organism is to the nanoparticle, thus indicating the effectiveness of the nanoparticles. Table 1 shows the inhibition zones (in mm) formed around the nanoparticles and positive controls (neomycin antibiotic and amphotericin B). The zones of inhibitions around the controls were ≥10 mm. These zones were larger than those formed around the test organisms, which confirms that the commercial antibiotic exhibits a higher growth inhibitory effect against both the Gram-positive and Gram-negative bacterial species examined in this study. The nanoparticles in this study were completely ineffective against the Gram-negative organisms *Escherichia coli*, *Salmonella enterica* and *Shigella sonnei*, but they were effective against Gram-positive *Staphylococcus aureus*. The zone of inhibition against *S. aureus* ranged from 3 mm at ratio 1:2 to 5 mm at ratio 1:4. The smaller-sized particles of ZnO with urea, ratio 1:4, effectively interacted more with bacterial membranes due to their large surface area, thus enhancing their antibacterial efficiency. A study conducted by Narayanan and Wilson (2012) showed the microbial activity of ZnO NPs to be higher against *S. aureus* than against *Escherichia coli* [42]. However, in a study by Albukhaty (2020), ZnO nanoparticles were found to be more effective against *E. coli* than *S. aureus* [8]. Emami-Karvani and Chehrazi (2012) conducted a similar study using disk diffusion and well diffusion agar methods [43].

Furthermore, the agar well diffusion results show that the microbial activity of ZnO NPs is higher against *S. aureus* than against *Escherichia coli*, and the disk diffusion results show the opposite. For antifungal activity in the current study, ZnO NPs were not effective against *Candida albicans*. However, in previous studies conducted by Sharma and Ghose (2015) and Taufiq et al. (2018), the growth inhibition (1.5–11.4 mm) of *Candida albicans* by ZnO NPs was observed [44,45]. Agar well diffusion is a qualitative assay used in preliminary screening for the presence of bioactivity. However, it does not provide an accurate estimation of the effect of antimicrobial activity [46]. 

For this reason, minimal inhibitory concentration (MIC) was performed to provide a quantitative assessment of the antimicrobial activity of ZnO NPs against the selected test microorganisms. MIC is defined as the lowest concentration (mg·L^−1^) of an antimicrobial agent, which, under strictly controlled in vitro conditions, completely prevents visible growth of the test strain of an organism [47]. The redox indicator resazurin dye showed that active microbial cells reduced the non-fluorescent resazurin (blue) to the fluorescent resorufin (pink) [48]. Microbial cell growth induced a chemical reduction from aerobic respiration, thus producing observable color changes in the wells [49]. The MICs where ZnO NPs were more effective at inhibiting pathogenic organisms are shown in Table 2.

As can be seen from the results presented in Table 2, the microbial species in the present study were inhibited from growing at varying concentrations. Effective antimicrobial activity is depicted by low MIC values. In the current study, a low MIC (6.25 μg/mL) was observed for bacterial species *Shigella sonnei* and *Staphylococcus aureus* for the ZnO NP ratio 1:1. For yeast species *Candida albicans,* the lowest MIC was also 6.25 for the ZnO NP ratio 1:1. Positive controls, neomycin and amphotericin B were more effective in inhibiting microbial growth (MIC: 1.5623–3.125) than ZnO NPs (MIC: 6.25–25 μg/mL). The MIC results for commercial antibiotics correlate with the agar well diffusion assay results in the present study. The rise of multi-drug-resistant (MDR) strains has prompted a need for the development of a new class of antimicrobial agents [50]. Research on finding effective drug candidates against increasing MDR strains has focused on developing urea derivatives containing aryl moieties as potential antimicrobial agents [51,52]. Both studies have shown that urea derivatives exhibit antimicrobial activity against bacterial and fungal strains. An increase in urea concentration could increase the antimicrobial activity of an antimicrobial agent. However, in the present study, similarly to the findings obtained by Patil et al. (2019) [52], after screening against similar strains, antimicrobial activity was only observed against *S. aureus* [52]. 

ZnO NPs are effective, stable and safe for use as antimicrobial agents; thus, they do not cause harm to humans or animals [19,25]. The active antimicrobial activity of ZnO NPs results from a release of reactive oxygen species (ROS), such as hydrogen peroxide (H_2_O_2_), OH- (hydroxyl radicals) and O_2_^2−^ (peroxide), from the surface of ZnO. ROS cause oxidative stress by damaging cellular proteins, cell membranes and DNA; accumulating in microorganisms; and making contact with bacterial cells, causing changes in the microenvironment within the contact area of the organisms and particles. This contact damages and disorganizes the bacterial cells and releases antimicrobial ions (Zn^2+^) inside the cells, thereby causing toxicity in the bacterial cells. The ions can damage the cell membrane and enter the intracellular area [19,53]. The agar well diffusion and MIC results show that the microbial activity of ZnO NPs is higher against Gram-positive bacteria (*S. aureus*) than against Gram-negative bacteria. This can be attributed to microbial cell wall interactions with the nanoparticles. The lipopolysaccharide structure in Gram-negative bacteria could restrict the attachment of ZnO NPs and prevent the zinc ions from passing across the outer membrane. 

## 4. Conclusions

The increasing global population has resulted in increased anthropogenic activities that pollute the water environment, thus elevating the occurrence of waterborne pathogens. Pathogenic microorganisms have developed resistance to antimicrobial agents due to continuous exposure and misuse. Microbial disinfection in wastewater using metallic nanoparticles has recently gained attention in scientific research. This results from their cost effectiveness and safety profile. In the current study, ZnO NPs were successfully synthesized, characterized and tested for antimicrobial activity against selected waterborne pathogens. SEM and TEM images revealed that nanoparticles produced by the MH process exhibit uniform spherical shapes. The absorption spectrum of the white crystal ZnO colloids prepared by urea reduction showed a surface plasmon absorption band with a maximum of 320 nm, indicating the presence of spherical or roughly spherical ZnO nanoparticles. FTIR spectral studies confirmed that the binding of urea with zinc occurs through Zn–O stretching. The X-ray diffraction suggested the existence of a hexagonal ZnO wurtzite-type structure as per the JCPDS card used for indexing. The positive inhibitory activity of ZnO NPs was observed against all tested bacterial strains, as well as against *C. albicans*. In general, a high aspect ratio, a small particle size, multilevel porosity and a large surface area of ZnO NPs can enhance antimicrobial performance. The surface area of the metal oxide nanoparticles that comes into contact with bacterial cells is directly proportional to the extent of antimicrobial activity exhibited by the particles. This high antimicrobial activity can be very useful, especially against microorganisms that are resistant to conventional antimicrobials. Metallic nanoparticles have enormous potential in water treatment technology. They could improve microbial water quality in waste treatment plants. Proper and effective wastewater treatment will protect and improve the health of people and their communities.

## Figures and Tables

**Figure 1 molecules-27-03532-f001:**
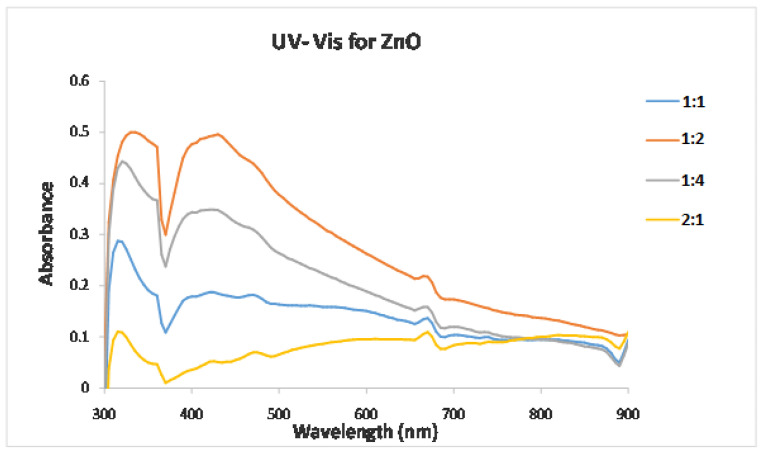
UV–Vis absorption spectrum recorded for ZnO NPs at different zinc acetate: urea ratios.

**Figure 2 molecules-27-03532-f002:**
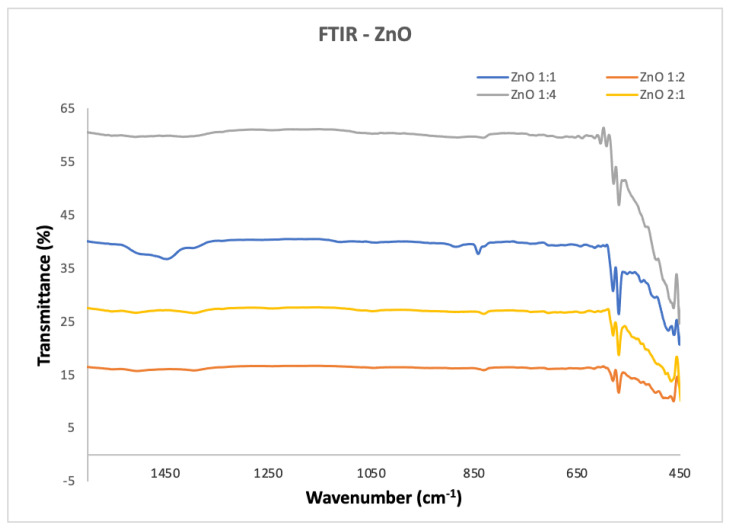
FTIR results for ZnO NPs at different zinc acetate: urea ratios.

**Figure 3 molecules-27-03532-f003:**
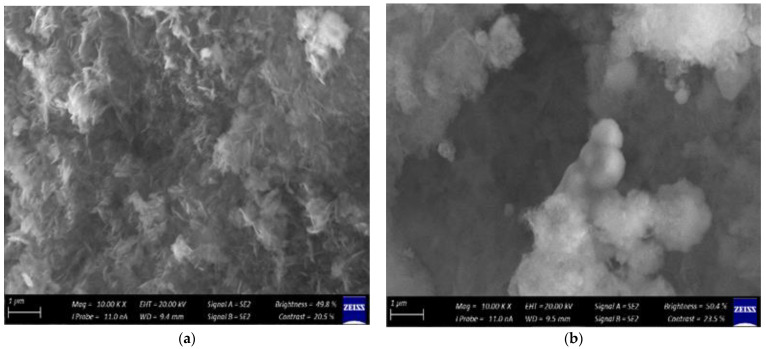
SEM micrographs for ZnO NPs at different zinc acetate: urea ratios: (**a**) 1:1, (**b**) 1:2, (**c**) 1:4 and (**d**) 2:1.

**Figure 4 molecules-27-03532-f004:**
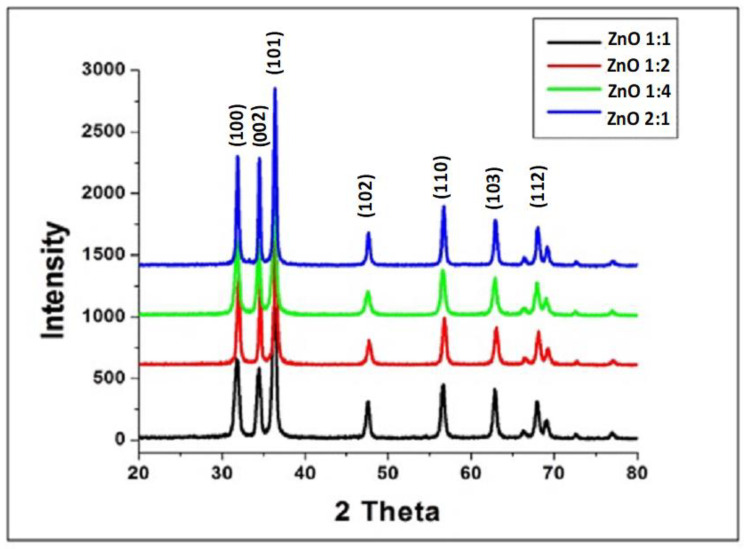
XRD results of ZnO NPs (zinc acetate: urea) after heating in a microwave oven for 4 min.

**Figure 5 molecules-27-03532-f005:**
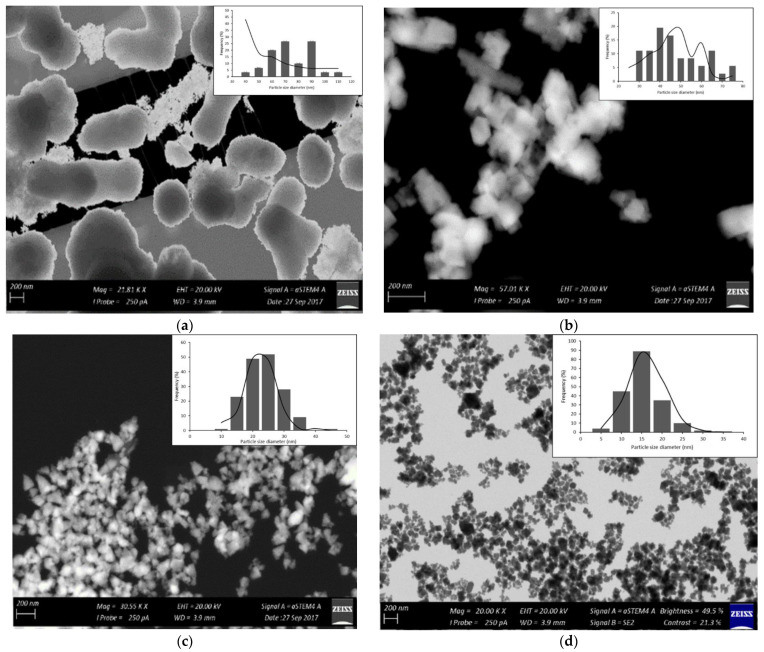
TEM micrographs for ZnO NPs at different zinc acetate: urea ratios: (**a**) 1:1, (**b**) 1:2, (**c**) 1:4 and (**d**) 2:1.

**Figure 6 molecules-27-03532-f006:**
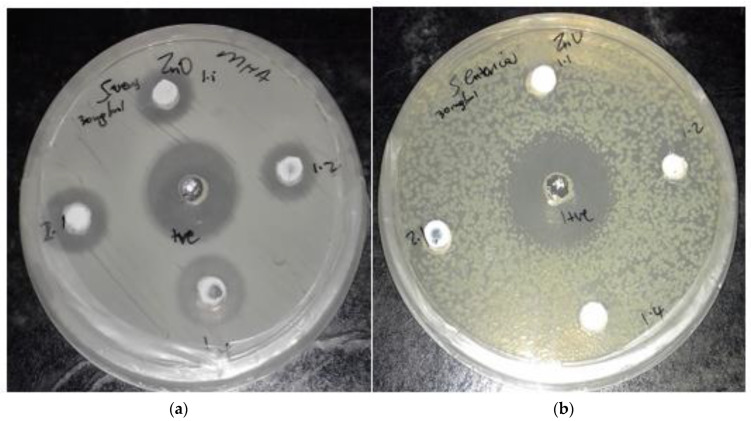
Clear zones of inhibition on MH agar plate on (**a**) Staphylococcus aureus and (**b**) Salmonella enterica.

**Table 1 molecules-27-03532-t001:** Mean zones of inhibition (in mm) produced by synthesized zinc oxide nanoparticles on the test organisms.

Microorganisms	Agar Well Diffusion Method				
	ZnO NPs (mm)				Antibiotic (Neomycin/Amphotericin B) (mm)
	1:1	1:2	1:4	2:1	
*Staphylococcus aureus*	4	3	5	4	10
*Escherichia coli*	0	0	0	0	10
*Salmonella enterica*	0	0	0	0	10
*Shigella sonnei*	0	0	0	0	13
*Candida albicans*	0	0	0	0	(Positive control measurement)

**Table 2 molecules-27-03532-t002:** Minimum inhibitory concentrations of bacterial and yeast species.

Microorganisms	Minimum Inhibitory Concentrations (MICs)				
	ZnO NPs				Antibiotic (Neomycin/Amphotericin B)
	1:1	1:2	1:4	2:1	
*Staphylococcus aureus*	6.25	6.25	12.5	12.5	3.125
*Escherichia coli*	25	25	25	25	1.5623
*Salmonella enterica*	12.5	12.5	12.5	25	1.5623
*Shigella sonnei*	6.25	12.5	12.5	12.5	1.5623
*Candida albicans*	6.25	12.5	12.5	12.5	3.125

## Data Availability

Not applicable.

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
