# Peer review of "Synthesis, Characterization and Antimicrobial Activity of Zinc Oxide Nanoparticles against Selected Waterborne Bacterial and Yeast Pathogens"

_molecules, 2022, doi:10.3390/molecules27113532_

Round 1
Reviewer 1 Report
The following issues must be resolved:
- Authors mentioned that the nanocatalyst was prepared using microwave heating crystallization technique. But there is no data about microwave heating conditions and instrument (microwave oven). Please add them to the manuscript.
- Please add an explanation about the control of temperature under microwave heating to the text. The experimental section must be improved and should be clear.
- It is recommended that authors should be included a small paragraph related to the importance of ZnO nanoparticles with citing to the following references: a) Materials Science and Engineering: B 2007, 139 (2-3), 265-269 b) The Scientific World Journal, vol. 2014, Article ID 925494, 8 pages, 2014.; c) Bioinorganic Chemistry and Applications, vol. 2018, Article ID 1062562, 18 pages, 2018; d) Resource-Efficient Technologies, 2017, 3(4), pp.406-413; e) Catalysis Communications 2008, 9 (2), 299-306; f) Nanomicro Lett. 2015;7(3):219-242,..
- What is the effect of urea concentration on size, morphology and antimicrobial activity of ZnO nanoparticles? Please explain in detail in the manuscript.
- What is the effect of microwave heating conditions (time, temperature and power of instrument) on size and morphology of ZnO nanoparticles? Please explain in detail in the manuscript.
- Please insert antimicrobial activity of the synthesized ZnO nanoparticle in a Table and compare its activity quantitatively with a standard and reference compound.
- Please add “(Zinc acetate: Urea)” to all figure captions.
- In the text explain how and which peaks were used for particle size measurements (X-Ray Diffraction).
Author Response
Dear Reviewer 1
All the corrections on the revised manuscript and the justifications to reviewers' comments have been done in Red font color.
The following issues must be resolved:
1. Authors mentioned that the nanocatalyst was prepared using microwave heating crystallization technique. But there is no data about microwave heating conditions and instrument (microwave oven). Please add them to the manuscript.
It has been done; Under the Materials and methods section, paragraph 1; page 3
2. Please add an explanation about the control of temperature under microwave heating to the text. The experimental section must be improved and should be clear.
Under the Materials and methods section, paragraph 1; page 3 was added to explain the control of temperature under microwave heating.
3. It is recommended that authors should be included a small paragraph related to the importance of ZnO nanoparticles with citing to the following references: a) Materials Science and Engineering: B 2007, 139 (2-3), 265-269 b) The Scientific World Journal, vol. 2014, Article ID 925494, 8 pages, 2014.; c) Bioinorganic Chemistry and Applications, vol. 2018, Article ID 1062562, 18 pages, 2018; d) Resource-Efficient Technologies, 2017, 3(4), pp.406-413; e) Catalysis Communications 2008, 9 (2), 299-306; f) Nanomicro Lett. 2015;7(3):219-242,..
Under the introduction, a paragraph (paragraph 7) was added to address the importance of ZnO nanoparticles. Most of the suggested authors are cited.
4. What is the effect of urea concentration on size, morphology and antimicrobial activity of ZnO nanoparticles? Please explain in detail in the manuscript.
Page 9, last paragraph addressed the importance of the effect of urea concentration on size, morphology and antimicrobial activity of ZnO nanoparticles.
5. What is the effect of microwave heating conditions (time, temperature and power of instrument) on size and morphology of ZnO nanoparticles? Please explain in detail in the manuscript.
The average particle diameters were 90 nm and 85 nm for samples treated by the MH process for 2 and 8 min, respectively. This means that as the heating time increase the particle size decrease.
The effect of microwave heating temperature and power were not investigated
6. Please insert antimicrobial activity of the synthesized ZnO nanoparticle in a Table and compare its activity quantitatively with a standard and reference compound.
Table 1shows the diameters of zones of inhibitions for nanoparticles and the controls. This is clearly stated under 3.3 Evaluation of the antimicrobial activity, Paragraph 1, Line 6.I have rephrased the sentence to make it clearer. On the same paragraph, Line 8-12, the comparisons of the diameters and the interpretation of results for well diffusion techniques is provided.
Table 2shows the Minimum Inhibitory Concentrations (MIC) of nanoparticles and controls. This is clearly stated under 3.3 Evaluation of the antimicrobial activity, Paragraph 4, Line 1-2. On the same paragraph, Line 2-7, the comparisons between MICs and the interpretation of results is provided.
7. Please add “(Zinc acetate: Urea)” to all figure captions.
“(Zinc acetate: Urea)” has been added to all figure captions.
8. In the text explain how and which peaks were used for particle size measurements (X-Ray Diffraction).
The most intense bands (or peaks) were used for particle size measurements.
All the observed peaks (100), (002), (101), (102), (110), (103), and (112) can easily be identified in Bragg’s reflectionassociated with the hexagonal wurtzite phase of ZnO. The peak broadening in XRD patterns clearly indicates the presence of small crystallites in the samples. The lattice constants a and c, of all nanoparticles, as calculated from the peak position of (100) and (002) planes, respectively; Page 7.
Reviewer 2 Report
In this work, zinc oxide (ZnO) nanoparticles (NPs) were synthesized using microwave heating. The purpose of this study is to use ZnO NPs against selected waterborne pathogenic microbes.
Please find below my comments and suggestions:
- Abstract. The term 'metallic NMs' is only cited in the abstract and NMs does not appear along the document. Perhaps you can define it or use the full word and omit the term NMs.
- Experimental Section. Please check the following aspects: i) brand of UV-visible spectrometer; ii) FTIR spectroscopy, what is the number of scans?; iii) SEM, brand of the equipment, sample preparation, etc; iv) X-Ray difraction, please include Scherrer equation; v) TEM brand of the equipment.
- Results and discussion: i) UV-visible, in the case of the plot, probably the use of crosses as marks to indicate the position of the axex at 400 nm, 500 nm, etc... ii) check the legend in UV-visible plot (is it 1.01; 1:02; 1:04 and 2:01 or 1:1; 1:2; 1.4 and 2.1?? iii) check legend in FTIR plot and compare with others. Please use the samne nomenclature; also I recommended a table with the band assignment; iv) In Figure 3, says "SEM results ofr , I would recommend "SEM micrographs"; v) In fIgure 5, there is no scale bar. This comment also applies for TEM experiments; v) in TEM, there is no maximum assignement according to JCRP; vi) particle size distribution from TEM micrograpjv) in XRD there is no assignment to the peaks.
Taking into consideration my comments, I recommend revision of the manuscript with major revion. I consider that it can be considered for publication after fixing some details.
Author Response
Dear Reviewer,
All the corrections on the revised manuscript and the justifications to your comments have been done in Red font color.
Reviewer 2
In this work, zinc oxide (ZnO) nanoparticles (NPs) were synthesized using microwave heating. The purpose of this study is to use ZnO NPs against selected waterborne pathogenic microbes.
Taking into consideration my comments, I recommend revision of the manuscript with major revion. I consider that it can be considered for publication after fixing some details.
Please find below my comments and suggestions:
- Abstract. The term 'metallic NMs' is only cited in the abstract and NMs does not appear along the document. Perhaps you can define it or use the full word and omit the term NMs.
“NMs” has been removed in the abstract and replaced with “nanoparticles”
Experimental Section. Please check the following aspects:
- i) brand of UV-visible spectrometer;
it has been added in the text : (T80+ UV-Vis spectrometer)
- ii) FTIR spectroscopy, what is the number of scans?
The number of scans was 16; it has been added to the text.
- iii) SEM, brand of the equipment, sample preparation, etc;
Vega 3 Tescan Scanning electron microscopywas used to determine the surface morphologies of the nanomaterials synthesized. For analysis, a few mg of each sample was placed on a carbon tape. The samples were then gold coated in order to make their surfaces conductive.
- iv) X-Ray diffraction, please include Scherrer equation;
The samples were analysed on an X’Pert Pro XRD with a Co tube. The phases were identified using X’Pert Highscore plus software, PAN ICSD and an ICDD database.
The average crystallite size of the most intense XRD peak (the dominant phase) was calculated using the Scherrer equation (1) below
D is the average crystallite size, k is a constant equal to 0.9; λ is the wavelength
(nm) of copper Kα radiation and its value is 0.154 nm; β, the FWHW of the peak of
interest obtained by XRD; Theta (θ) is the Bragg angle.
- v) TEM brand of the equipment.
Transmission electron microscopy (TEM), JEOL - JEM – 2100 electronmicroscope, was employed.
Results and discussion:
- i) UV-visible, in the case of the plot, probably the use of crosses as marks to indicate the position of the axex at 400 nm, 500 nm, etc...
On the axis “major ticks” have been added to indicate the position of axis at 400 nm , 500 nm, etc…
- ii) check the legend in UV-visible plot (is it 1.01; 1:02; 1:04 and 2:01 or 1:1; 1:2; 1.4 and 2.1??
The legend has been corrected
- iii) check legend in FTIR plot and compare with others. Please use the same nomenclature; also I recommended a table with the band assignment;
The band assignments are already stated in the text (page 5), adding a table for the band assignment again will be like a repetition.
- iv) In Figure 3, says "SEM results ofr , I would recommend "SEM micrographs";
It has been changed to “SEM micrographs”
- v) In figure 5, there is no scale bar. This comment also applies for TEM experiments;
New figures with scale bar have been added
- v) in TEM, there is no maximum assignment according to JCRP;
It has been corrected, see Page 8 and 9 last paragraph.
- vi) particle size distribution from TEM micrograph
Particle size distribution has been conducted, see Page 8.
- jv) in XRD there is no assignment to the peaks.
XRD pattern of the ZnO nanopowder at different ratios revealed that the sample under study exhibited high purity as demonstrated by the single-phase with no extra peaks.
This means that all the peaks observed after XRD analysis correspond to ZnO.
The peaks were assigned.
Round 2
Reviewer 1 Report
Dear Editor
The paper was revised according to the reviewer’ comments.
In its current state it is ready for publication in your journal.
Best regards